# Fast-ELECTRA for Efficient Pre-training

**Chengyu Dong**[1][*][†] **Liyuan Liu**[2] **Hao Cheng**[2] **Jingbo Shang**[1] **Jianfeng Gao**[2] **Xiaodong Liu**[2][†]
[1]University of California, San Diego    [2]Microsoft Research
{cdong, jshang}@ucsd.edu  {lucliu, chehao, jfgao, xiaodl}@microsoft.com

## Abstract

ELECTRA pre-trains language models by detecting tokens in a sequence that have been replaced by an auxiliary model. Although ELECTRA offers a significant boost in efficiency, its potential is constrained by the training cost brought by the auxiliary model. Notably, this model, which is jointly trained with the main model, only serves to assist the training of the main model and is discarded post-training. This results in a substantial amount of training cost being expended in vain. To mitigate this issue, we propose Fast-ELECTRA, which leverages an existing language model as the auxiliary model. To construct a learning curriculum for the main model, we smooth its output distribution via temperature scaling following a descending schedule. Our approach rivals the performance of state-of-the-art ELECTRA-style pre-training methods, while significantly eliminating the computation and memory cost brought by the joint training of the auxiliary model. Our method also reduces the sensitivity to hyper-parameters and enhances the pre-training stability.

## 1 Introduction

ELECTRA (Clark et al., 2020) is a pre-training method that trains the models to predict whether each token is the original token or synthetically generated replacement, in a corrupted input sequence. The token replacement is sampled from a distribution of possible tokens given the context, which is typically the output distribution of a masked language model. Henceforth, we will refer to this masked language model as the auxiliary model [1]. This pre-training task, known as *replaced token detection* (RTD), has shown great advantages in training and data efficiency compared to other pre-training tasks such as *masked language modeling* (MLM) (Devlin et al., 2019). ELECTRA-style pre-training and its variations have been increasingly popular in advancing natural language understanding capabilities (Meng et al., 2021; Chi et al., 2021; Meng et al., 2022; He et al., 2021; Bajaj et al., 2022).

Despite its effectiveness, one pitfall of ELECTRA that restricts its popularity is the design choices regarding how the auxiliary model is jointly trained with the main language model. This is originally intended to provide a natural curriculum on the RTD task for the main model's learning since the auxiliary model will start off weak and gradually gets better, and thus progressively ramp up the difficulty of the token replacements through pre-training (Clark et al., 2020). However, this design results in a substantial amount of resources (including computation and memory) being wasted, since the auxiliary model will be discarded after each training round. This issue becomes more severe considering that the training cost of the auxiliary model scales with the training cost of the main model, in terms of both the model size and training updates. This issue becomes even more severe considering the practical difficulty of balancing the auxiliary and main model's optimizations (Meng et al., 2022), which often demands multiple training rounds to search for the optimal hyper-parameter.

In this work, we propose a simple ELECTRA-style pre-training alternative that can greatly alleviate this issue. In specific, we employ an existing language model as the auxiliary model, which can either be retrieved from a public repository or from a previous training round. However, directly

---

[*]Work was done during an internship at Microsoft Research.

[†]Correspondence to: Chengyu Dong <cdong@ucsd.edu>, Xiaodong Liu <xiaodl@microsoft.com>

[1]In previous works (Clark et al., 2020), the auxiliary model is also referred to as the generator while the main model is referred to as the discriminator.

pre-training with this existing language model impairs the performance, potentially because the auxiliary model generates token replacements that are too difficult. To reduce the difficulty of the token replacements, we smooth the output distribution of the auxiliary model by temperature scaling and gradually decrease the temperature through pre-training following a pre-defined descending schedule. Such a strategy effectively constructs a curriculum for the main model's learning without the need to jointly train the auxiliary model.

We have developed a method called Fast-ELECTRA that further enhances the efficiency of ELECTRA, a method already known for its competitive efficiency improvements. In addition, Fast-ELECTRA achieves comparable performance to state-of-the-art ELECTRA-style pre-training methods in various pre-training settings. As shown in Figure 1, Fast-ELECTRA achieves the same downstream performance [2] with a BERT-base equivalent model (Devlin et al., 2019) using consistently less training than original ELECTRA, and cut the overall training time by more than 50 hours from a total of 200 hours. Based on more extensive calculations and experiments, Fast-ELECTRA can reduce the computation cost of the auxiliary model by 67% and the overall computation cost of ELECTRA by about 20-25%. Fast-ELECTRA can also reduce the memory cost of the auxiliary model by 97% and the overall memory cost by 10-20%. Furthermore, Fast-ELECTRA reduces the sensitivity of ELECTRA-style pre-training to the hyper-parameter choice, especially those responsible for the delicate balance between the optimizations of the main model and the auxiliary model. Fast-ELECTRA also improves the training stability of ELECTRA-style pre-training, thus being more promising for scaling up to large language models.

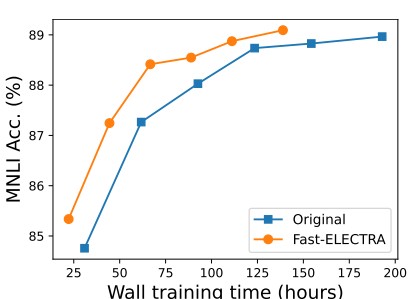

Figure 1: Downstream task performance (MNLI accuracy, Avg m/mm) versus wall-clock training time for BERT-base equivalent models pre-trained with the "Original" ELECTRA design and our "Fast-ELECTRA". Training time is measured on a node with 8 Tesla V100 GPUs. More experiment details can be found in Section 4.1.

The rest of the paper will be organized as follows. In Section 2 we will briefly introduce the background of ELECTRA-style pre-training and its efficiency advantage compared to other pre-training methods. In Section 3 we will motivate and present our method in detail. In Section 4 we will conduct extensive experiments to test the effectiveness, efficiency, and robustness of our method. In Section 5 we will demonstrate the necessity of the auxiliary model by comparing with model-free construction of RTD tasks, and the necessity of a learning curriculum by comparing with pre-training with a fixed auxiliary model.

## 2 PRELIMINARIES

**Masked Language Modeling.** The MLM task used in BERT (Devlin et al., 2019) trains the language model to predict randomly masked tokens in a sequence. Specifically, given an input sequence $\boldsymbol{x} = [x_1, x_2, \cdots, x_n]$, MLM generates a *masked sequence* $\boldsymbol{x}_{\text{masked}}$ by randomly selecting a few tokens at positions $\boldsymbol{m} = [m_1, m_2, \cdots, m_K]$ replace them with [mask]. The model is then trained to predict the original tokens at the masked positions. The training objective is

$$L_{\text{MLM}}(\theta) = \mathbb{E}_{\boldsymbol{x}} \sum_{i \in \boldsymbol{m}} - \log p_\theta(x_i | i, \boldsymbol{x}_{\text{masked}}),$$

where $p_\theta(\cdot | i, \boldsymbol{x}_{\text{masked}})$ is the output distribution of the model $\theta$ at position $i$, conditioned on the masked sequence $\boldsymbol{x}_{\text{masked}}$.

**ELECTRA-style Pre-training.** Unlike MLM, ELECTRA uses RTD as the objective which pre-trains the language model to detect replaced tokens in a sequence. Specifically, given an input sequence $\boldsymbol{x}$, ELECTRA generates a *corrupted sequence* $\boldsymbol{x}_{\text{corrupted}}$ by randomly selecting a few positions $\boldsymbol{m}$ and replace the token $x_i$ at each position $i \in \boldsymbol{m}$ with a corresponding token $\hat{x}_i$ that is likely semantically similar but not necessarily the same. We will refer to $\hat{x}$ as the *replaced token*,

---

[2]Following previous works (He et al., 2021; Bajaj et al., 2022), we use the results on MNLI (Williams et al., 2017) to indicate the performance on downstream tasks.

which is sampled from a probability distribution over the entire vocabulary. The language model is then trained to predict whether each token in $\boldsymbol{x}_{\text{corrupt}}$ is the original token or a replacement, namely

$$L_{\text{RTD}}(\theta) = \mathbb{E}_{\boldsymbol{x}} \sum_{i=1}^{n} -1(\hat{x}_i \neq x_i) \log p_\theta(i, \boldsymbol{x}_{\text{corrupt}}) - 1(\hat{x}_i = x_i) \log(1 - p_\theta(i, \boldsymbol{x}_{\text{corrupt}})),$$

where $p_\theta(i, \boldsymbol{x}_{\text{corrupt}})$ is the probability of replacement predicted by the model at position $i$, and $1(\cdot)$ is the indicator function.

ELECTRA-style pre-training has greatly improved the training efficiency compared to MLM and has dominated the state-the-of-arts on natural language understanding benchmarks (He et al., 2021; Meng et al., 2022; Bajaj et al., 2022). For example, ELECTRA can rival the downstream performance of RoBERTa (Liu et al., 2019b), a competitive MLM-style pre-training method, with only $25\%$ of the computation cost (Clark et al., 2020).

**Auxiliary Model and Joint-training.** The pivot of the RTD task in ELECTRA-style pre-training is the probability distribution which the replaced tokens are sampled from, which is typically determined by an auxiliary masked language model. Specifically, to generate the corrupted sequence $\boldsymbol{x}_{\text{corrupted}}$ mentioned above, ELECTRA first replaces all the tokens at positions $\boldsymbol{m}$ with [mask] and obtain a masked sequence $\boldsymbol{x}_{\text{masked}}$. The probability distribution of the replaced token at each position is then simply the corresponding output distribution of the auxiliary model evaluated on $\boldsymbol{x}_{\text{masked}}$, namely $\hat{x}_i \sim p_{\text{aux}}(\cdot|i, \boldsymbol{x}_{\text{masked}})$ for each $i \in \boldsymbol{m}$.

In the original ELECTRA design, the auxiliary model is jointly trained with the main model, which is shown to be necessary for the effectiveness of the pre-training (Clark et al., 2020). The overall training objective is defined as $\min_{\theta,\theta_{\text{aux}}} L_{\text{MLM}}(\theta_{\text{aux}}) + \lambda L_{\text{RTD}}(\theta)$, where $\theta_{\text{aux}}$ refers to the parameters of the auxiliary model, and $\lambda$ is a hyper-parameter that balances the optimizations of the auxiliary model and the main model. The intuition here is that joint training can provide a natural curriculum on the RTD task for the main model's learning since the difficulty of the replaced tokens will gradually increase, as the auxiliary model improves throughout pre-training.

## 3 COMPUTATION OVERHEAD REDUCTION OF AUXILIARY MODEL

**Computing and Memory Cost of the Auxiliary Model.** Despite that joint training of the auxiliary model is effective, it results in a significant amount of computation resources being wasted since the auxiliary model will be discarded once the training is finished. Although the auxiliary model is typically smaller than the main model (about $1/4$ - $1/3$ of the depth (Bajaj et al., 2022)), its size has to be scaled with the main model (*e.g.*, the same width (Bajaj et al., 2022)) to maintain the difficulty of the RTD task relative to the capacity of the main model, which means more computation resources will be wasted in training larger language models.

Our estimations show that, in each training trial, in each training update, and for each input batch, the auxiliary model expends about $67\%$ of the computation cost (about 4.0e11 FLOPs) of the main model when pre-training a BERT-base equivalent model. The overall computation cost scales dramatically as one pre-trains with larger batch size, for more updates, and more hyper-parameter searching rounds. The auxiliary model also consumes a significant amount of memory during training, which is about $30\%$ of the memory cost (about 7 GB) of the main model when pre-training a BERT-base equivalent model. The excessive memory cost of ELECTRA could in turn induce more computation cost than necessary because only smaller batch sizes can fit into the memory.

**Fast-ELECTRA.** We propose a simple alternative to significantly reduce the computation and memory cost of the auxiliary model inspired by simulated annealing (Kirkpatrick et al., 1983). Specifically, we employ an existing language model as the auxiliary model, which can either be retrieved from a public repository or from a previous training experiment. To reduce the difficulty of the RTD task, we leverage temperature scaling to smooth the output distribution of this existing model. The replaced tokens in the RTD task are then sampled from its smoothed output distribution, namely

$$\hat{x}_i \sim \text{Softmax}\left(\frac{\log p_{\text{aux}}(\cdot|i, \boldsymbol{x}_{\text{masked}})}{T}\right), \tag{1}$$

To create a learning curriculum for the main model similar to the effect of joint training, we simply schedule the temperature $T$ by an exponential decay function during pre-training, namely

$$T = 1 + (T_0 - 1) \cdot \exp(-u/\tau). \tag{2}$$

Here $u$ denotes the fraction of the training updates, while $T_0$ and $\tau$ are two hyper-parameters that control the initial temperature and decay rate respectively. Since now the auxiliary model is not jointly trained, the training objective is simply $\min_\theta L_{\text{RTD}}(\theta)$.

The main advantage of Fast-ELECTRA is training efficiency as our auxiliary model is only used for inference during pre-training. The computation cost of the auxiliary model is now only $1/3$ of the original, while the memory cost of the auxiliary model is now only about $1/30$ of the original. With offline preprocessing, we can further reduce both the computation and memory cost of the auxiliary model during pre-training to $0$ (see more details in Section 4.3).

Our design also improves ELECTRA's robustness to the hyper-parameter settings (see Section 4.4) and its training stability (see Section 4.5).

# 4 EXPERIMENTS

## 4.1 EXPERIMENT SETUP

**Pre-training Setup.**    We conduct experiments with two standard settings, *Base* and *Large*, following previous works (Devlin et al., 2019; Meng et al., 2021; Bajaj et al., 2022). Specifically, we employ Wikipedia and BookCorpus (Zhu et al., 2015) (16 GB of texts, 256M samples) for pre-training with a sequence length of $512$. We use a cased sentence piece BPE vocabulary of $128$K tokens following (He et al., 2020), since larger vocabulary size improves LLMs without significant additional training and inference cost (Bao et al., 2020). We conduct pre-training for $125$K updates with a batch size of $2048$. For Fast-ELECTRA, the auxiliary model is pre-trained following a standard MLM style with a learning rate of 5e-4.

**Model Architecture.**    Our main model (discriminator) in the Base setting follows the BERT$_{\text{base}}$ architecture (Devlin et al., 2019), namely a $12$-layer transformer with $768$ hidden dimensions plus T5 relative position encoding (Raffel et al., 2019) with $32$ bins. We employ Admin (Liu et al., 2020a; 2021) for model initialization to stabilize the training. Our main model in the Large setting follows BERT$_{\text{Large}}$, namely a $24$-layer transformer with $1024$ hidden dimensions and $128$ relative position encoding bins. We follow previous works (Clark et al., 2020; Bajaj et al., 2022) to set the size of the auxiliary model (generator), namely $4$ layers for the Base setting and $6$ layers for the large setting. More details of the model configuration are listed in Table 4.

**Downstream evaluation setup.**    We conduct evaluation on downstream tasks following the setup in previous works (Meng et al., 2021; Bajaj et al., 2022). Specifically, we evaluate on GLUE (Wang et al., 2018) language understanding benchmark with a single-task, single-model fine-tuning setting following previous works. We report Spearman correlation on STS-B, Matthews correlation on CoLA, and accuracy on the rest of the datasets. We follow the training hyperparameters suggested by Liu et al. (2019a; 2020b), such as the use of an AdaMax optimizer (Kingma & Ba, 2015b). Detailed hyperparameter settings can be found in Appendix A.

**Baselines.**    We compare our method with various baselines with an experiment setup same as ours, in terms of dataset, model size, and computation cost. We also incorporate baselines with similar experiment setup to allow more comparisons. For example, in the Large setting, we report baselines that pre-train for 1M updates but with a bach size of $256$, which aligns our setup in terms of the total number of processed tokens. We obtain results of these baselines from their papers and follow-up works, whichever are higher. We also reimplement METRO as our baselines. Both the re-implemented METRO (Bajaj et al., 2022) and our method are implemented within the same codebase, which is built on top of FAIRSEQ (Ott et al., 2019), a popular open-sourced package.

**Hyper-parameter Settings.**    We follow previous works (Clark et al., 2020; Bajaj et al., 2022) to select the generator size, namely $4$ layers for the Base setting and $6$ layers for the Large setting. For our re-implemented METRO, we set the loss weight as $70$ and $50$ for the Base and Large setting respectively while the learning rate as 5e-4 for both settings, which produces the best results in our experiments. For Fast-ELECTRA, we set the initial temperature as $2$, the decay rate as $0.1$, and the

Table 1: Results on GLUE development set. "-" indicates that no public reports are available. "†" indicates the model is pre-trained for 1M updates with batch size of 256. "‡" indicates the model is pre-trained for 100K updates with batch size of 8K.

| Model | MNLI-(m/mm) (Acc.) | QQP (Acc.) | QNLI (Acc.) | SST-2 (Acc.) | CoLA (Mat. Corr.) | RTE (Acc.) | MRPC (Acc.) | STS-B (Spear. Corr.) | Average Score |
|---|---|---|---|---|---|---|---|---|---|
| **Base Setting** | | | | | | | | | |
| BERT (Devlin et al., 2019) | 84.5/ - | 91.3 | 91.7 | 93.2 | 58.9 | 68.6 | 87.3 | 89.5 | 83.1 |
| RoBERTa (Liu et al., 2019b) | 85.8/85.5 | 91.3 | 92.0 | 93.7 | 60.1 | 68.2 | 87.3 | 88.5 | 83.3 |
| XLNet (Yang et al., 2019) | 85.8/85.4 | - | - | 92.7 | - | - | - | - | - |
| DeBERTa (He et al., 2020) | 86.3/86.2 | - | - | - | - | - | - | - | - |
| TUPE (Ke et al., 2020) | 86.2/86.2 | 91.3 | 92.2 | 93.3 | 63.6 | 73.6 | 89.9 | 89.2 | 84.9 |
| ELECTRA (Clark et al., 2020) | 86.9/86.7 | 91.9 | 92.6 | 93.6 | 66.2 | 75.1 | 88.2 | 89.7 | 85.5 |
| MC-BERT (Xu et al., 2020) | 85.7/85.2 | 89.7 | 91.3 | 92.3 | 62.1 | 75.0 | 86.0 | 88.0 | 83.7 |
| COCO-LM (Meng et al., 2021) | 88.5/88.3 | 92.0 | 93.1 | 93.2 | 63.9 | 84.8 | 91.4 | 90.3 | 87.2 |
| AMOS (Meng et al., 2022) | 88.9/88.7 | **92.3** | 93.6 | 94.2 | **70.7** | **86.6** | 90.9 | **91.6** | 88.6 |
| DeBERTaV3 (He et al., 2021) | **89.3/89.0** | - | - | - | - | - | - | - | - |
| METRO (Bajaj et al., 2022) | 89.0/88.8 | 92.2 | 93.4 | **95.0** | 70.6 | 86.5 | 91.2 | 91.2 | 88.6 |
| METRO_ReImp | 89.0/88.9 | 92.0 | 93.4 | 94.4 | 70.1 | 86.3 | **91.4** | 91.2 | 88.5 |
| Fast-ELECTRA | 89.4/88.8 | 92.1 | **93.8** | 94.5 | **71.4** | 85.6 | **91.4** | **91.6** | **88.7** |
| **Large Setting** | | | | | | | | | |
| BERT† | 86.6/ - | - | - | - | - | - | - | - | - |
| RoBERTa‡ | 89.0/ - | 91.9 | 93.9 | **95.3** | 66.3 | 84.5 | 90.2 | 91.6 | 87.8 |
| XLNet† | 88.4/ - | 91.8 | 93.9 | 94.4 | 65.2 | 81.2 | 90.0 | 91.1 | 87.0 |
| TUPE† | 88.2/88.2 | 91.7 | 93.6 | 95.0 | 67.5 | 81.7 | 90.1 | 90.7 | 87.3 |
| METRO_ReImp | 89.9/90.2 | **92.5** | **94.5** | 94.3 | 69.7 | **88.8** | **91.9** | 91.6 | 89.2 |
| Fast-ELECTRA | **90.1/90.2** | 92.4 | **94.5** | 95.1 | **72.1** | 87.4 | 90.7 | **91.9** | **89.3** |

learning rate as 1e-3 for both the Base and Large settings. Detailed hyper-parameter settings are elaborated in Appendix A.

## 4.2 DOWNSTREAM PERFORMANCE

Table 1 lists the downstream evaluation results of Fast-ELECTRA and competitive baselines under the Base and Large setting. Fast-ELECTRA matches previous state-of-the-arts with jointly-trained generator, in terms of the overall GLUE score and results on more reliable datasets such as MNLI.

## 4.3 TRAINING EFFICIENCY

Our approach can significantly improve the training efficiency of ELECTRA-style pre-training, in terms of both computation and memory cost.

**Computation Cost.** The computation cost of ELECTRA is contributed by both the main model and the auxiliary model. We assume the backward propagation has approximately twice the computation cost of the forward propagation following previous works (Kaplan et al., 2020). Therefore, the computation cost of the auxiliary model is reduced by $2/3$ since it is only used for inference in Fast-ELECTRA.

We estimate the computation cost more accurately by calculating training FLOPs for each input batch in each training update (*i.e.*, one forward plus one backward propagation). We follow the formula introduced by (Hoffmann et al., 2022) to calculate the FLOPs for both the main model and the auxiliary model. As shown in Table 2, our method can reduce the overall computation cost by about 20-25% for both base and large models.

**Memory cost.** The memory cost of ELECTRA is contributed by both the main model and the auxiliary model. Considering standard training setup of language models such as the Adam optimizer (Kingma & Ba, 2015a) and mixed-precision training (Micikevicius et al., 2017), the memory cost of each trainable parameter is contributed by its weight, its gradient, and its corresponding state buffers in the optimizer, which requires 20 bytes in total (Smith et al., 2022). In contrast, for a parameter that is only used in inference, the memory cost consists of only its weight, namely 2 bytes. Therefore, the memory cost of the auxiliary model is significantly reduced since it is only used for inference in Fast-ELECTRA.

Besides the model parameters, the intermediate activations of the computation graph stored for backward propagation consume significant memory as well. In typical implementations, one can employ checkpointing (Gruslys et al., 2016; Chen et al., 2016) to trade computation for memory and reduce the memory footprint within each encoding layer. Nevertheless, the activations after each encoding layer still need to be stored (Smith et al., 2022). Therefore, the activation memory

Table 2: Computation cost per batch per update and memory consumption of ELECTRA. Note that for the original ELECTRA, we exclude the embedding layer from the memory calculation of the auxiliary model since it is shared between the auxiliary model and the main model. We also report the computation and memory cost on realistic computation infrastructures in Appendix B.1.

| Model | Method | Computation (GFLOPs) | | | Memory (GB) | | |
|---|---|---|---|---|---|---|---|
| | | Main | Auxiliary | Total | Main | Auxiliary | Total |
| Base | Original | 591.9 | 398.6 | 990.5 | 23.0 | 7.0 | 30.0 |
| | Fast-ELECTRA | 591.9 | 132.9 | 724.8 | 23.0 | 0.25 | 23.3 |
| | Ratio | 1.0 | 0.33 | 0.73 | 1.0 | 0.04 | 0.77 |
| Large | Original | 1407.7 | 653.9 | 2061.6 | 60.2 | 14.4 | 74.6 |
| | Fast-ELECTRA | 1407.7 | 218.0 | 1625.6 | 60.2 | 0.42 | 60.6 |
| | Ratio | 1.0 | 0.33 | 0.79 | 1.0 | 0.03 | 0.81 |

scales with the number of layers, which means the auxiliary model always consumes about $1/4$-$1/3$ additional memory on top of the main model. The activation memory also scales with the sequence length, number of hidden dimensions, and batch sizes. In contrast, the auxiliary model in Fast-ELECTRA consumes $0$ intermediate activation memory since it is used only for inference.

We report the estimated memory cost[3] in Table 2. Our method can reduce the memory cost of the auxiliary model by about $97\%$ and the overall memory cost by more than $20\%$ for both base and large models.

**Offline preprocessing for $0$ auxiliary computation and memory cost.** Last but not least, Fast-ELECTRA also makes the offline preprocessing technique feasible for ELECTRA-style pre-training, which can further reduce both the computation and memory cost of the auxiliary model to $0$ during pre-training. In specific, one can generate and dump the training data of the RTD task (*i.e.*, the corrupted input sequence and the binary target indicating whether each token is replaced or not) at each epoch before pre-training, since the only varying parameter in our auxiliary model is the temperature and it can be determined by the epoch number. These dumped training data can be reused in subsequent pre-training rounds, for example, for hyper-parameter search or continual pre-training. This strategy would be particularly preferred for large-scale pre-training.

Note that a similar strategy used in MLM pre-training is called static masking (Devlin et al., 2019; Liu et al., 2019b). But Fast-ELECTRA equipped with offline preprocessing can be even more efficient than MLM pre-training with static masking since the training targets are binary instead of dependent on the vocabulary size.

### 4.4 ROBUSTNESS TO THE HYPER-PARAMETER SETTINGS

In this section, we investigate the robustness of our method to the hyper-parameter settings. We are mostly interested in hyper-parameters that control the learning curriculum, namely the difficulty of the RTD task throughout pre-training, since it is the key difference between Fast-ELECTRA and the original ELECTRA and is also important to the pre-training's effectiveness.

In practice, we would strongly prefer an ELECTRA-style pre-training method that is friendly to the learning curriculum tuning. This is because it is challenging to control the learning curriculum since a positive curriculum in the short term is not necessarily beneficial to the entire training course. Searching for the optimal curriculum often requires training till the end. Controlling the learning curriculum is particularly challenging for pre-training because the effectiveness of a curriculum can only be revealed by the performance on representative downstream tasks, which requires time-consuming fine-tuning. There is no reliable and instant signal in pre-training that can indicate the effect of a curriculum.

**Size of the auxiliary model.** The size of the auxiliary model can directly affect the learning curriculum. A large auxiliary model can converge faster (Arora et al., 2018), thus creating a more difficult RTD task for the main model. Previous works have observed that an auxiliary model that is too large or too small can both damage the effectiveness of the pre-training (Clark et al., 2020). However, tuning the size of the auxiliary model in practice can be overly problematic, as the model

---

[3]For the activation memory, we account for the entire batch size and ignore gradient accumulation here as it depends on the specific GPU memory size.

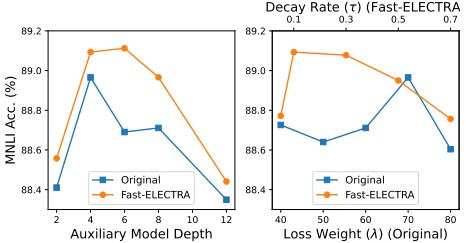
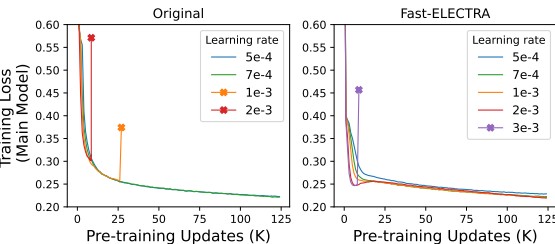

Figure 2: Downstream task performance (MNLI accuracy, Avg m/mm) versus the depth of the auxiliary model (Left) and the hyper-parameters (Right) used in the "Original" ELECTRA or "Fast-ELECTRA".

Figure 3: Training loss curves of the main model when pre-training with the original ELECTRA (*Left*) and Fast-ELECTRA (*Right*). "×" indicates the divergence of the training.

architecture changes in each training attempt, which means the experiment or hardware setup very likely needs to be re-configured to maintain the training efficiency.

We show that compared to the original ELECTRA, Fast-ELECTRA is more robust to the size of the auxiliary model, thus being more friendly to use in practice. We experiment on the original ELECTRA and Fast-ELECTRA with multiple auxiliary model depths, while for other hyper-parameters (*e.g.*, loss weight $\lambda$ in the original ELECTRA and decay rate $\tau$ in Fast-ELECTRA), we select the ones that produce the best performance for each depth respectively. As shown in Figure 2 left, Fast-ELECTRA can achieve more predictable performances as the auxiliary model depth varies.

**Curriculum Schedule.** We also experiment on hyper-parameters provided by each method that can directly control the learning curriculum when the auxiliary model is determined. In the original ELECTRA, it is mainly the loss weight $\lambda$ that controls the learning curriculum, since it balances the optimizations of the auxiliary model and the main model. We neglect the learning rate here since it affects the auxiliary model and the main model simultaneously. In Fast-ELECTRA, the decay rate $\tau$ is used to control the learning curriculum. We neglect the initial temperature here since the default value of 2 works well across different auxiliary model and main model settings.

To fairly compare the sensitivities of the performance to $\lambda$ and $\tau$ given their different scales, we sweep each hyper-parameter around two of its best values under different settings. For example, we modulate $\lambda$ in [40, 50, 60, 70, 80], since 70 and 50 are the best values we found for the Base and Large settings respectively. We modulate $\tau$ in [0.05, 0.1, 0.3, 0.5, 0.7] since 0.1 and 0.5 are the best values we found for a 4-layer auxiliary model and a 12-layer auxiliary model respectively.

Figure 2 right shows the downstream performances obtained by experimenting with these hyper-parameter values. It can be seen that the performance changes abruptly when $\lambda$ varies in the original ELECTRA. In contrast, Fast-ELECTRA produces a smoother performance curve as $\tau$ varies.

### 4.5 TRAINING STABILITY

In this section, we study the training stability, which is known to be a major bottleneck in training large language models (Liu et al., 2020a). Upon scaling up the model size, one may need to reduce the learning rate, scale down the variance of the weight initialization, apply heavy gradient clipping, or re-configure the model architecture (Bajaj et al., 2022; Smith et al., 2022). However, these remedies often sacrifice the efficiency and/or effectiveness of the pre-training (Bajaj et al., 2022).

Although we cannot afford large-scale experiments, we make an initial attempt to inspect the training stability with a standard model size (BERT-base equivalent). In specific, we conduct pre-training with excessively large learning rates. Figure 3 shows that the original ELECTRA quickly diverges as the learning rate increases, while pre-training with Fast-ELECTRA can remain stable with a learning rate as large as 2e-3, almost triple the maximum value allowed by the original method.

## 5 ABLATION STUDIES

### 5.1 ELECTRA-STYLE PRE-TRAINING FREE OF AUXILIARY MODELS

Here, we attempt to construct an RTD task without an auxiliary model. As mentioned in Section 2, the pivot of RTD task is the probability distribution which the replaced tokens are sampled from, which

we will refer to as the replaced token distribution. We experiment with the following alternatives to define the replaced token distribution without an auxiliary model.

- *Uniform*: a uniform distribution defined over the entire vocabulary.
- *Term frequency*: a distribution where the probability mass of a token is equal to its frequency across the entire training corpus.
- *Smoothed one-hot*: a distribution where the probability mass of the correct token is $1 - \alpha$, while the probability mass of any other token is $\alpha/(|\mathcal{V}| - 1)$. Here $\mathcal{V}$ indicates the vocabulary and $\alpha = 0.35$ is the typical prediction error rate of an auxiliary model on the masked tokens.

Figure 4 visualizes the above distributions for an example sequence [4]. Note that the term frequency approximately follows Zipf's law (Newman, 2004). We also plot the replaced token distribution produced by an auxiliary model for reference.

We also include learning curriculums defined by these distributions by interpolation, as intuitively smoothed one-hot is more difficult than other distributions since it can be viewed as an auxiliary model that always predicts the original token correctly with high confidence (See more details in Appendix B.3).

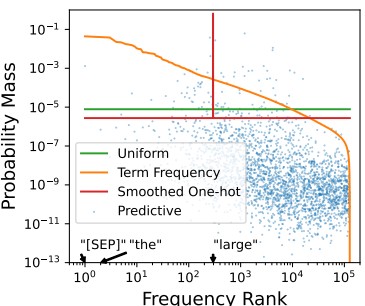

Figure 4: The probability distribution of the replaced token in an example sequence defined by various methods. The token classes are ranked by the term frequency in the $x$-axis.

Table 3 shows the downstream performance when pre-training with these replaced token distributions [5]. Pre-training with the uniform distribution diverges early at around 10K updates. We suspect this is because the replaced tokens sampled from a uniform distribution are too easy to detect since most of the tokens would be rare and unique considering a vast vocabulary. This is supported by our observation that the training loss of the discriminator plunges to 0 at the very beginning of the pre-training, after which the training diverges. In contrast, interpolating the uniform distribution with the more difficult smoothed one-hot distribution can converge smoothly. In fact, such an interpolation also yields better downstream performance than smoothed one-hot alone, which can be attributed to the benefit of the learning curriculum.

Interestingly, pre-training with term frequency alone can converge and yield reasonable downstream performance ($\sim 85.7\%$ MNLI Acc.), whereas interpolating term frequency and smoothed one-hot leads to consistently worse downstream performance than term frequency alone. We suspect that term frequency is superior because it better indicates the difficulty of possible replaced tokens in a context.

Finally, we note that pre-training with these auxiliary-model-free replaced token distributions are still inferior to auxiliary-model-based ones in terms of the downstream performance upon convergence.

Table 3: Downstream task performance (MNLI accuracy, average m/mm) when pre-training with different replaced tokens distributions.

| Method | Fast-ELECTRA | Uniform | Term Frequency | Smoothed One-hot | UNF → SOH | TF → SOH |
|---|---|---|---|---|---|---|
| MNLI Acc. | 89.1 | Diverged | 85.7 | 83.8 | 85.3 | 84.9 |

## 5.2 DOES THE CURRICULUM MATTER FOR ELECTRA-STYLE PRE-TRAINING?

In this section, we investigate whether a learning curriculum, namely a gradually more difficult RTD task, is necessary for ELECTRA-style pre-training. We focus on the case when an auxiliary model is available, given the inferior performance of model-free alternatives as mentioned above. The comparative experiment here is *fixed-auxiliary pre-training*, where the replaced token distribution is simply the output distribution of a pre-trained and fixed auxiliary model, without any modification.

---

[4]A quote from Clark et al. (2020), "*most current training methods require [large] amounts of compute to be effective*", where "*[*]*" indicates the token to be replaced.

[5]We use UNF → SOH and TF → SOH to denote uniform-to-smoothed one-hot interpolation and term frequency-to-smoothed one-hot interpolation respectively.

Figure 5 (left) shows the downstream of pre-training with a fixed auxiliary model. In our experiments, for more than 10K training updates from the beginning, the main model's replaced token detection accuracy remains 0. Yet with more training updates it starts to converge and yield decent downstream performance ($\sim 88.3\%$ on MNLI).

Nevertheless, an appropriate curriculum greatly improves training efficiency. Compared to fixed-auxiliary training, temperature-scaling this same auxiliary model improves the accuracy from $80\%$ to $85\%$ on MNLI when the number of training updates is limited (e.g., 2K). An appropriate curriculum can also advance the downstream performance upon convergence ($88.4\%$ vs. $89.1\%$).

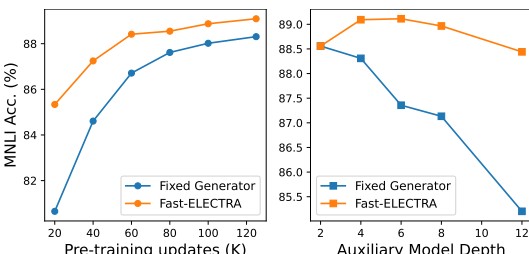

Figure 5: (*Left*): Downstream performance (MNLI accuracy, Avg m/mm) at multiple intermediate checkpoints, obtained by pre-training with a fixed auxiliary model and Fast-ELECTRA. (*Right*): Downstream performance at the last checkpoint versus the depth of the auxiliary model, obtained by pre-training with a fixed auxiliary model and Fast-ELECTRA.

Finally, we found that a learning curriculum is particularly important for auxiliary models with a large capacity. As shown in Figure 5 right, temperature-scaling the auxiliary model improves the MNLI accuracy for more than $3\%$ compared to fixed-auxiliary training. This implies learning curriculum alleviates the sensitivity of the pre-training effectiveness to the auxiliary model's capacity.

In Appendix B.2, we further experiment on alternative ways such as Dropout to design a learning curriculum. We also experiment on alternative functions such as stepwise decay to schedule the curriculum. These alternatives can yield decent downstream performance, albeit slightly worse than the default settings in Fast-ELECTRA.

## 6 RELATED WORK

**Variations of ELECTRA-style Pre-training.** Here we briefly summarize the variations of the ELECTRA-style pre-training in the literature. Xu et al. (2020) pre-trains the model to predict the original token from a small candidate set, instead of predicting a binary target. Meng et al. (2021) introduces two additional training objectives including the prediction of the original token and alignment between corrupted sequences from the same source. Hao et al. (2021) learns to sample more difficult replace tokens. Meng et al. (2022) automatically constructs a difficult learning signal by an adversarial mixture of multiple auxiliary models. He et al. (2021) argues that embedding sharing between the main model and the auxiliary model may hurt pre-training and proposes a stop gradient operation during back-propagation. Bajaj et al. (2022) conducts a comprehensive ablation study on ELECTRA and highlights several important improvements such as large vocabulary size and relative position embedding, and successfully scales ELECTRA-style pre-training up to billions of parameters. Zhang et al. (2022) observes the existence of "false negative" replaced tokens, namely those that are not exactly the same but are synonyms to the original ones, and proposes to correct them by synonym look-up and token similarity regularization.

## 7 CONCLUSION

In this work, we focus on the training cost of the auxiliary model in ELECTRA-style pre-training and propose a simple method that employs existing language models and annealed temperature scaling to greatly alleviate the issue. Our method achieves comparable performance to state-of-the-art while being more efficient and robust. In general, our approach empowers ELECTRA-style pre-training with more flexibility, opening up potential applications in continual learning, transfer learning, and knowledge distillation for language models.

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

## A  HYPER-PARAMETER SETTINGS

We follow the standard practice in previous works to set the hyper-parameters (Devlin et al., 2019; Meng et al., 2021; Bajaj et al., 2022). For MLM pre-training of the generator, we fix the mask ratio as $15\%$. When sampling sequences for pre-training, we respect document boundaries and avoid concatenating texts from different documents. We did not mask special tokens follow the standard BERT practice. We conduct pre-training on NVIDIA Tesla V100 with 32GB memory and fine-tuning on NVIDIA Tesla P100 with 16GB memory. Table 5 lists the detailed hyper-parameter settings for pre-training. Table 6 lists the detailed hyper-parameters used for fine-tuning.

Table 4: Configuration of model architectures. Note that the auxiliary model has the same configuration in each encoding layer as the main model, albeit having fewer layers. As a reference for the calculation of memory cost, we list the embedding layer separately since it is shared between the main model and the auxiliary model in the original ELECTRA design.

| Model | Depth (Main) | Depth (Aux) | Hidden Size | FFN Width | Attention Heads | # Params (Main) | # Params (Aux) | # Params (Embed) |
|-------|------|------|------|------|------|------|------|------|
| Base  | 12 | 4 | 768  | 3072 | 12 | 184 M | 127M | 98M  |
| Large | 24 | 6 | 1024 | 4096 | 16 | 434 M | 208M | 131M |

Table 5: Hyperparameter settings for pre-training.

| Hyperparameters | Base | Large |
|-----------------|------|-------|
| Max Steps | 125K | 125K |
| Optimizer | Adam | Adam |
| Peak Learning Rate (Fast-ELECTRA) | $1 \times 10^{-3}$ | $1 \times 10^{-3}$ |
| Peak Learning Rate (METRO$_{\mathrm{ReImp}}$) | $5 \times 10^{-4}$ | $5 \times 10^{-4}$ |
| Loss Weight (METRO$_{\mathrm{ReImp}}$) | 70 | 50 |
| Batch Size | 2048 | 2048 |
| Warm-Up Steps | 10K | 10K |
| Sequence Length | 512 | 512 |
| Vocabulary Size | 128K | 128K |
| Relative Position Encoding Buckets | 32 | 128 |
| Relative Position Encoding Max Distance | 128 | 256 |
| Adam $\epsilon$ | 1e$-$6 | 1e$-$6 |
| Adam $(\beta_1, \beta_2)$ | $(0.9, 0.98)$ | $(0.9, 0.98)$ |
| Clip Norm | 2.0 | 2.0 |
| Dropout | 0.1 | 0.1 |
| Weight Decay | 0.01 | 0.01 |

## B  ADDITIONAL EXPERIMENTS

### B.1  TRAINING EFFICIENCY

We test the overall computation cost and memory cost on specific computation infrastructures, as hardware-centered optimization can be critical to training efficiency (Rasley et al., 2020). We conduct experiments on two typical infrastructures, including a node with $4\times$ GeForce RTX 3090 GPUs (24GB memory each, w/o NVLink), and a node with $8\times$ Tesla V100 GPUs (32GB memory each, w/o NVLink). We measure the computation cost by the wall time (in seconds) per training update (SPU), and the memory cost by the peak memory occupied by all tensors on one GPU throughout training, averaged over all GPUs.

As shown in Table 7, Fast-ELECTRA can reduce the overall computation and memory cost of ELECTRA-style pre-training across different computation infrastructures and model settings consis-

Table 6: Hyperparameter search space in fine-tuning.

| Hyperparameters | Base | Large |
|---|---|---|
| Sequence Length | 256 | 256 |
| Optimizer | AdaMax | AdaMax |
| Peak Learning Rate | {5e-5,1e-4, 3e-4} | {5e-5,1e-4, 3e-4} |
| Max Epochs | {2,3,5,10} | {2,3,5,10} |
| Batch size | {16, 32, 64} | {16, 32, 64} |
| Learning rate decay | Linear | Linear |
| Weight Decay | {0, 0.01} | {0, 0.01} |
| Warm-up Proportion | {6 %, 10 %} | {6 %, 10 %} |
| Adam $\epsilon$ | 1e-6 | 1e-6 |
| Adam $(\beta_1, \beta_2)$ | $(0.9, 0.98)$ | $(0.9, 0.98)$ |
| Gradient Clipping | 1.0 | 1.0 |
| Dropout | 0.1 | 0.1 |

Table 7: Computation cost and memory consumption of ELECTRA measured on specific infrastructures

| Model | Method | $4\times$ RTX 3090 | | $8\times$ Tesla V100 | |
|---|---|---|---|---|---|
| | | Computation (SPU) | Memory (GB) | Computation (SPU) | Memory (GB) |
| Base | Original | 10.0 | 13.7 | 5.6 | 13.7 |
| | Fast-ELECTRA | 7.7 | 11.6 | 4.0 | 11.4 |
| | Ratio | 0.77 | 0.84 | 0.71 | 0.83 |
| Large | Original | 13.8 | 19.1 | 6.7 | 19.1 |
| | Fast-ELECTRA | 11.2 | 16.2 | 5.3 | 16.2 |
| | Ratio | 0.81 | 0.85 | 0.79 | 0.84 |

tently[6]. The reduction of computation cost matches our calculation in Section 4.3, while the reduction of memory cost is slightly less than that from our calculation, which is due to gradient accumulation that reduces the peak memory.

## B.2    Alternative Curriculum Designs for ELECTRA-style Pre-training

In this section, we explore alternative ways to design the learning curriculum when an auxiliary model is available. In general, a model-based curriculum can be determined by two functions, namely the schedule function and the augmentation function.

**Augmentation function.**    An augmentation function is used to reduce the difficulty of the replaced token task generated by an existing auxiliary model. In Fast-ELECTRA, we utilized temperature scaling as an augmentation function to smooth the output distribution of the auxiliary model. Yet, essentially any method that can change the model's output distribution can be utilized as an augmentation function. Possible methods include changing the model's output distribution directly, changing the behaviors of the model weights or modules, or changing the model's input sequence. Here we consider the following alternative augmentation functions.

- *Logarithmic interpolation*: Interpolate the auxiliary model's output distribution with an easy distribution such as a uniform distribution or term frequency. Since interpolating with a uniform distribution is in fact equivalent to temperature scaling [7], we only consider the interpolation

---

[6]Note that Fast-ELECTRA can in fact support a larger batch size per GPU and thus potentially speed up training further, due to reduced memory cost. Nevertheless, in our tests, we set the batch size per GPU to be the same for the original ELECTRA and Fast-ELECTRA, such that the computation reduction of Fast-ELECTRA involves no contribution of less memory cost.

[7]Logarithmic interpolation with a uniform distribution is $\log p := \gamma \log p_{\text{UNF}} + (1-\gamma) \log p_\theta = -\gamma \log |\mathcal{V}| + (1 - \gamma) \log p_\theta$, while temperature scaling can be formulated as $\log p := (1/T) \log p_\theta$. Therefore, these two will be the same up to a constant difference, which will be canceled after applying Softmax on the log probabilities.

with term frequency, namely $p_{\text{TF}\to\theta}(\cdot|i, \boldsymbol{x}_{\text{masked}}) := p_{\text{TF}}^{\gamma} \odot p_{\theta}^{1-\gamma}(\cdot|i, \boldsymbol{x}_{\text{masked}})$, where $\odot$ denotes element-wise product [8].

- *Dropout*: Enable all activation dropout layers in the auxiliary model, and set all their drop rates as $\gamma$.

- *Drop attention*: Enable all attention dropout layers in the auxiliary model, and set all their drop rates as $\gamma$.

- *Drop token*: Randomly replace $\gamma$ fraction of tokens (except special tokens such as "`[MASK]`") in the auxiliary model's input sequence with "`[UNK]`", namely the token representing unknown tokens.

Figure 6 left shows the downstream performance achieved by pre-training with these augmentation functions. Here for each augmentation, we use an exponentially decayed function to schedule $\gamma$ similar to Equation 2, and search its best hyper-parameters (*i.e.*, $\gamma_{\max}$ and $\tau$) based on the final performance. One may find that all these augmentation functions can achieve decent downstream performance. Nevertheless, in practice, we prefer temperature scaling as it is easy to implement and achieves slightly better performance.

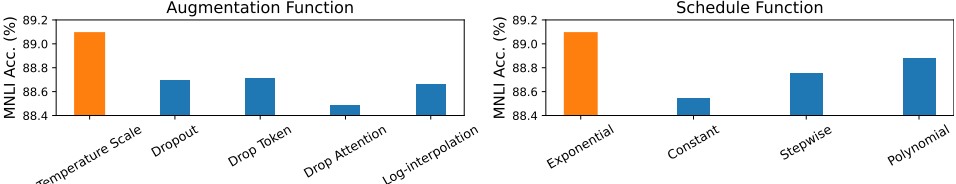

Figure 6: Downstream performance (MNLI accuracy, Avg m/mm) obtained by pre-training with different augmentation functions (*Left*) and different schedule functions (*Right*). The augmentation function and schedule function used in Fast-ELECTRA are highlighted.

**Schedule function.** A schedule function is used to determine the parameter of the augmentation function (*e.g.*, $\gamma$) at the $u$-th fraction of training updates, such that the difficulty of the replaced token detection task gradually increases through pre-training. In previous sections, we have experimented on an exponential decay function. Here we experiment on alternative schedule functions as follows.

- *Constant*: $\gamma(u) = \gamma_0$.

- *Polynomial decay*: $\gamma(u) = \gamma_{\max}(1 - u)^{\tau}$, where $\tau$ here controls the decay rate and a larger $\tau$ results in a faster decay.

- *Stepwise decay*: $\gamma(u) = \gamma_{\max}(1 - \lfloor u\tau \rfloor/\tau)$, where $\lfloor \cdot \rfloor$ denotes the floor function and $\tau$ determines the number of decays.

Figure 6 right shows the downstream performance achieved by temperature-scaling with its parameter scheduled by these functions. For each schedule function, we search for its best hyper-parameter based on the final performance. One can find that the constant schedule leads to significantly lower performance than exponential decay, while other schedules can achieve comparable performance.

### B.3 LEARNING CURRICULUM FREE OF AUXILIARY MODELS

We experiment on curriculum RTD tasks defined by replaced token distributions free of auxiliary models. As mentioned in Section 5.1, the smoothed one-hot distribution is intuitively more difficult than the uniform distribution and term frequency. Therefore, we can interpolate these distributions with a varied coefficient to gradually increase the difficulty of the RTD task during training. Concretely, we can define the replaced tokens distributions as

- Interpolation between uniform and smoothed one-hot (UNF→ SOH):

$$p_{\text{UNF}\to\text{SOH}} := p_{\text{UNF}}^{\gamma} \odot p_{\text{SOH}}^{1-\gamma}.$$

---

[8]We choose to logarithmically interpolate the distributions because we empirically observed it is slightly better than linearly interpolating.

- Interpolation between term frequency and smoothed one-hot (TF→ SOH):

$$p_{\text{TF}\rightarrow\text{SOH}} := p_{\text{TF}}^{\gamma} \odot p_{\text{SOH}}^{1-\gamma}.$$

Here $\odot$ denotes the element-wise multiplication, and $0 \leq \gamma \leq 1$ and is scheduled by an exponentially decayed function, similar to Equation 2. We choose to logarithmically interpolate the distributions because we empirically observed it is slightly better than linearly interpolating.

