# OpenReview forum: "Fast-ELECTRA for Efficient Pre-training"
_ICLR.cc/2024/Conference — ICLR 2024 poster_

### Official Review · Reviewer_QMNg · 2023-10-30

**Soundness:** 2 fair
**Presentation:** 2 fair
**Contribution:** 2 fair
**Rating:** 6
**Confidence:** 3

**Summary:**

This paper explores efficient ELECTRA training methods by advocating the use of a pre-trained/existing language model as an auxiliary model, rather than simultaneous training of the model and auxiliary model.

**Strengths:**

1. The method is both intuitive and effective.
2. The problem it tackles is highly practical.

**Weaknesses:**

1. The proposed method appears tailored specifically for ELECTRA, potentially limiting its applicability and community interest.
2. Could we consider applying a continual learning method (e.g., [1]) to enhance ELECTRA's efficiency?

[1]: Adapting a Language Model While Preserving its General Knowledge, Ke et al., EMNLP 2022

**Questions:**

See above

---

> ### Author Response · Authors · 2023-11-23
>
> Thank you for the valuable feedback. Please see our detailed response below.
>
>
> **___**
>
> __Q. Applicability of ELECTRA-style pretraining__
>
> We note that ELECTRA is widely applied in training encoding models and also among the state-of-the-art. Our method can further extend its applicability as it reduces the training cost and memory consumption of ELECTRA to be comparable to other pretraining schemes such as Mask Language Modeling (MLM).
>
> Furthermore, we firmly believe that the ELECTRA-style pretraining, which can achieve huge performance improvement over MLM with the same model size and training data, still has lots of merits that are worth studying, not only for the pretraining of encoding/decoding language models but also for the pretraining of large models in a wider context.
>
> **___**
>
> __Q. Correlation with continual learning__
>
> We appreciate the reviewer for bringing this work [1] to our attention. We believe there are indeed many correlations between ELECTRA-style pretraining and continal learning.
>
> On one side, existing continual learning methods can be great motivations for better designs of ELECTRA-style pretraining. Since strong auxiliary models can often hurt the performance of the main model in ELECTRA-style pretraining, it is necessary to design an auxiliary model training scheme that is aware of the main model learning. This requires the auxiliary model to be able to generate meaningful token replacements while not too difficult to ruin the main model learning. This is highly related to the idea proposed in the work mentioned by the reviewer, namely to adapt existing language models to both preserve necessary general knowledge and incorporate domain-specific knowledge.
>
> On the other side, ELECTRA-style pretraining may also be adapted for the continual learning field. For example, one may consider using a small auxiliary model trained on the specific domain to better adapt the main model while preserving necessary general knowledge.
>
> We will include the above discussion in the revision.
>
> **___**
>
> __Reference__
>
> [1] Adapting a Language Model While Preserving its General Knowledge, Ke et al., EMNLP 2022

---

### Official Review · Reviewer_rs11 · 2023-10-30

**Soundness:** 3 good
**Presentation:** 4 excellent
**Contribution:** 3 good
**Rating:** 6
**Confidence:** 4

**Summary:**

This paper proposes a simple and effective technique to improve electra training. By replacing the training of the auxiliary model with a pre-trained model together with temperature scaling and a gradually decreased temperature, the proposed method significantly reduced the memory usage and training time of electra training.

**Strengths:**

1. Simple and effective method.
2. Good performance.
3. Very clear presentation.

**Weaknesses:**

The scale of models in experiments seems a bit limited under the current standard. Have you tried larger models?

**Questions:**

See weakness.

---

> ### Author Response · Authors · 2023-11-23
>
> Thank you for the valuable feedback. Please see our detailed response below.
>
> **___**
>
> __Q. Larger models__
>
> We believe for encoding models, the BERT-base and BERT-large equivalent model sizes, as covered in this paper, are among the most widely used ones. Existing works have also mostly adopted such model sizes [1, 2].
>
> Nevertheless, we note that our method can extend the applicability of ELECTRA for training potential large-scale models, as it reduces the training cost and memory consumption of ELECTRA, such that it is now comparable to other popular pretraining schemes such as Mask Language Modeling (MLM).
>
> **___**
>
> __Reference__
>
> [1] RoBERTa: A robustly optimized BERT pretraining approach. Liu et al., 2019.
>
> [2] DeBERTaV3: Improving Deberta Using Electra-style Pre-training with Gradient-Disentangled Embedding Sharing. He et al., 2021.

---

### Official Review · Reviewer_dT26 · 2023-11-01

**Soundness:** 3 good
**Presentation:** 3 good
**Contribution:** 3 good
**Rating:** 6
**Confidence:** 4

**Summary:**

This paper proposes efficient pretraining with ELECTRA by using a fixed generator (auxiliary model) instead of training the aux model together with the discriminator for replace token detection (RTD). In order to simulate a curriculum of difficulty that training of aux model provides, it uses an exponentially decaying schedule on the *temperature* used to sample from aux model.

Efficiency: Since the aux model is fixed, Fast-ELECTRA saves on backward passes leading to overall 20-30% FLOPs per step. (This calculation ignores original training of the auxiliary model.) It also saves 20-25% memory in the aux model. One could also cache the aux model predictions and save a lot more FLOPs (30-40%).

Quality: Fast-ELECTRA is competitive with many BERT and ELECTRA related baselines, even slightly better on some GLUE downstream evals

Robustness: The paper evaluates robustness to curriculum schedule and discriminator model size, and finds that Fast-ELECTRA behaves more gracefully compared to ELECTRA and can handle higher learning rates.

The paper also performs interesting ablation studies with using some simple aux models and different (linear/poly) schedules for curriculum. Overall, the paper presents a conceptually interesting finding that that a fixed pretrained aux model can be used, with a temperature schedule. Incomplete comparisons to earlier ideas made it harder to judge novelty

**Strengths:**

Novelty: The idea of using a fixed aux model for efficiency is interesting, and novel to my knowledge (although I'm not entirely sure since I could not find much discussion about this). Similarly the idea of using decaying temperature as a curriculum in this context is quite interesting

Quality: The paper provides a nice analysis of computation and memory benefits of the method.

Clarity: The paper is easy to follow for most part. Connections to prior work and some other details could be presented better.

**Weaknesses:**

Comparison to prior work:

- It would be helpful to highlight the most relevant work in Table 1 that a reader should focus in. Additionally, is there any evaluation on prior work that uses fixed generator? It would also help to include some FLOPs comparison to the baselines used in Table 1. The paper will also help with a discussion on accuracy-efficiency tradeoff. Lack of such discussions made it harder to assess the full value of proposed method.

- Recent paper (Dong et al.) from ICML 2023 proposed a different strategy of decoupling generator and discriminator optimizers and has better GLUE metrics than Fast-ELECTRA. This does not dilute the contributions of this paper much because Fast-ELECTRA also leads to training speed up, and is conceptually different. However it will be helpful to include and compare to this method. It could be an interesting open question if the gap to Dong et al. can be reduced with a fixed aux model.


Missing discussions: Most of the paper assumes the existence of a good pretrained aux model, but ignores the cost of training the aux model itself. Does Fast-ELECTRA truly reduce total FLOPs in that case?


Overall I believe that the paper makes a positive contribution. I'm currently assigning a score of weak accept mainly due to above reservations about comparisons to prior work.

Dong et al. Understand and Modularize Generator Optimization in ELECTRA-style Pretraining. ICML 2023

**Questions:**

- In Eq 2 what is the range of $u$? What is the final temperature in that case? If $u$ is indeed fraction of training updates, then at the max value of $u=1$ the final temperature has a value different from 1. Is that intended?

- Section 3 says “the auxiliary model expends about 67% of the computation cost” If aux model is just 1/3 the size, why does it contribute so much to computation cost?

---

> ### Author Response · Authors · 2023-11-23
>
> Thank you for the valuable feedback. Please see our detailed response below.
>
> **___**
>
> __Q. Efficiency improvement over existing methods__
>
> Among Table 1 of our paper, those methods that follow the ELECTRA-style pretraining framework are the most relevant to this paper, including MC-BERT [1], COCO-LM [2], AMOS [3], DeBERTaV3 [4], and METRO [5]. For a brief introduction to these methods, one can refer to the related work (Section 6) in our paper.
>
> Nevertheless, note that all these works train the generator jointly with the discriminator. To the best of our knowledge, there is no prior work using a fixed generator, potentially because it will fail to match the performance achieved by joint training, as also shown by Section 5.2 in our paper.
>
> Therefore, we believe the major contribution of our proposed method, namely using a fixed generator for ELECTRA-style pretraining without significant performance loss, is **orthogonal** to prior works. Existing ELECTRA-style pretraining methods can be modified with a fixed generator and achieve similar efficiency improvement.
>
> Finally, regarding the accuracy-efficiency trade-off, we didn’t observe significant performance loss by using our method to improve the training efficiency. Our method is still among the state-of-the-art in terms of downstream performance, as shown in Table 1 of our paper. Nevertheless, it may be indeed slightly worse compared to the most recent work mentioned by the reviewer, as discussed below.
>
> **___**
>
> __Q. Comparison to a recent work__
>
> We appreciate the reviewer for bringing up this recent work [6]. We note that our paper is different from [6] in terms of both the motivation and the methodology.
>
> In terms of motivation, [6] aims to improve the effectiveness of the ELECTRA-style pretraining, by controlling the training trajectory of the generator for better discriminator learning. In comparison, our work aims to improve the training efficiency of the ELECTRA-style pretraining, by reducing the training cost of the generator.
>
> In terms of the methodology, [6] decouples the optimizers of the generator and the discriminator, while the generator is still jointly trained with the discriminator. In contrast, Fast-ELECTRA completely decouples the generator and discriminator training. As a result, the generator can be trained separately, and only needs to be trained once and reused for the discriminator training.
>
> Finally, we note that there is indeed a gap between the performance of our method and that of [6]. We agree with the reviewer that it would be tempting if such a gap could be reduced while the auxiliary model is still fixed, such that we can achieve efficiency and performance at the same time. We believe this is possible by designing a better learning curriculum but we would like to leave it as a future work.
>
> We will include the above discussion in the revision per the reviewer’s suggestion.
>
>
>
> **___**
>
> **__Q. Training cost of the auxiliary model__**
>
> We note that Fast-ELECTRA will reduce the overall training cost even if one pre-trains the auxiliary model themself.
>
> First, the practical development of language models often requires multiple rounds of trial and error to locate the best architecture setup and hyperparameter combinations. With the joint training design in the original ELECTRA, the auxiliary model will be trained and discarded repeatedly in each round, resulting in significant resources being wasted. In contrast, with our method, one only needs to train the auxiliary model once and can reuse it for subsequent training rounds of the main model.
>
> Furthermore, we note that Fast-ELECTRA can reduce the memory cost of ELECTRA-style pretraining as well, which also improves the training efficiency. In fact, large-scale language model pretraining, calling for larger model sizes and longer sequence lengths, is probably more often bottlenecked by memory consumption [7]. With the joint training design in the original ELECTRA, the auxiliary model will consume significant memory in addition to that of the main model, which limits the applicability of ELECTRA-style pretraining in many memory-constraint scenarios compared to other pretraining frameworks such as Masked Language Modeling (MLM). In contrast, with our method, the auxiliary model’s memory cost during main model training can be significantly reduced or even eliminated (with offline processing), which can potentially make ELECTRA more memory-efficient than MLM, and in turn, can also reduce the training time as one can now fit more training data into the memory.

---

> > ### Author Response · Authors · 2023-11-23
> >
> > **___**
> >
> > **__Q. Final temperature of the temperature schedule__**
> >
> > We would like to clarify that the final temperature would indeed be slightly larger than 1 given the current schedule design. More specifically, since our default decay rate is set as 0.1, and the final fraction of training updates is 1, the final temperature would be 1.0000453, which is quite close to 1.
> >
> > Note that this is not intended but rather for the simplicity of the schedule design. It is possible to enforce the final temperature to be 1, but would require some non-trivial modification to the exponential decay function, which we believe wouldn’t significantly affect the results.
> >
> >
> >
> > **___**
> >
> > **__Q. Computation cost of the auxiliary model compared to the main model__**
> >
> > We note that the computation cost of the language model consists of both the cost of the Transformer backbone and that of the embedding layers. In terms of the Transformer backbone, the computation cost of the auxiliary model is indeed about 1/3 of that of the main model since it is about 1/3 of the size. However, in terms of the embedding layers, the computation cost of the auxiliary model is in fact same as that of the main model, since the computation cost of the embedding layer (including both input \& classification head) can be approximated as
> >
> > *Cost (embedding) = 2 $\times$ Sequence length $\times$ Embedding size $\times$ Vocabulary size*
> >
> > where the sequence length, embedding size, and vocabulary size are the same between the auxiliary and the main model.
> >
> > For more details, we decompose the computation costs of the auxiliary and the main model for a BERT-base equivalent model as follows. One can see that the total computation cost of the auxiliary model is indeed about 67% of that of the main model.
> >
> > | Computation cost (GFLOPs) | Transformer backbone | Embedding | Total |
> > | ------------------------- | -------------------- | --------- | ----- |
> > | Main                      | 289.8                | 302.1     | 591.9 |
> > | Auxiliary                 | 96.6                 | 302.1     | 398.6 |
> > | Auxiliary / Main          | 0.33                 | 1.0       | 0.67  |
> >
> > **___**
> >
> > __Reference__
> >
> > [1] MC-BERT: Efficient Language Pre-training via a Meta Controller. Xu et al., 2020.
> >
> > [2] COCO-LM: Correcting and contrasting text sequences for language model pretraining. Meng et al., 2021.
> >
> > [3] Pretraining Text Encoders with Adversarial Mixture of Training Signal Generators. Meng et al., 2022.
> >
> > [4] DeBERTaV3: Improving Deberta Using Electra-style Pre-training with Gradient-Disentangled Embedding Sharing. He et al., 2021.
> >
> > [5] METRO: Efficient Denoising Pretraining of Large Scale Autoencoding Language Models with Model Generated Signals. Bajaj et al., 2022.
> >
> > [6] Understand and Modularize Generator Optimization in ELECTRA-style Pretraining. Dong et al., 2023.
> >
> > [7] Using DeepSpeed and Megatron to Train Megatron-Turing NLG 530B, A Large-Scale Generative Language Model. Smith et al., 2022.

---

### Meta-Review · Area_Chair_iNzU · 2023-12-11

**Metareview:**

This paper studies faster training for Electra. The key idea is to use fixed generator instead of training the auxiliary model usually trained together with the discriminator for replace token detection in Electra. The paper shows that this leads to significantly reduction in the computation and memory cost of joint training in Electra.

**Justification For Why Not Higher Score:**

The approach is relatively simple (which is not necessarily a bad thing) but the approach of using pretrained networks has been tried several times in different settings. Thus, the novelty seems somewhat limited.

**Justification For Why Not Lower Score:**

Simulating the necessary curriculum with temperature schedule is simple but neat. Their ablations on role of curriculum, robustness to large lr were also useful.

---

### Decision · Program_Chairs · 2024-01-16

Accept (poster)